# Psychometric Properties of the WHO-5 Well-Being Index among Nurses during the COVID-19 Pandemic: A Cross-Sectional Study in Three Countries

**DOI:** 10.3390/ijerph191610106

**Published:** 2022-08-16

**Authors:** Mariela Loreto Lara-Cabrera, Moisés Betancort, Amparo Muñoz-Rubilar, Natalia Rodríguez-Novo, Ottar Bjerkeset, Carlos De las Cuevas

**Affiliations:** 1Department of Mental Health, Faculty of Medicine and Health Sciences, Norwegian University of Science and Technology (NTNU), 7091 Trondheim, Norway; 2Nidelv Community Mental Health Centre, Division of Psychiatry, St Olavs Hospital Trondheim University Hospital, 7006 Trondheim, Norway; 3Department of Clinical Psychology, Psychobiology and Methodology, Universidad de La Laguna, 38200 San Cristóbal de La Laguna, Spain; moibemo@ull.edu.es; 4Faculty of Health Sciences, Universidad Central de Chile, Santiago 8370178, Chile; cristina.munoz@ucentral.cl; 5Departament of Nursing, Universidad de La Laguna, 38200 San Cristóbal de La Laguna, Spain; nrodrigu@ull.edu.es; 6Faculty of Nursing and Health Sciences, Nord University, Levanger Campus, 8026 Levanger, Norway; ottar.bjerkeset@nord.no; 7Department of Internal Medicine, Dermatology and Psychiatry, Universidad de La Laguna, 38200 San Cristóbal de La Laguna, Spain; ccuevas@ull.edu.es; 8Instituto Universitario de Neurociencia (IUNE), Universidad de La Laguna, 38200 San Cristóbal de La Laguna, Spain

**Keywords:** COVID-19, cross-cultural, item response theory, measurement invariance, mental health, nursing staff, pandemics, public health, validation, WHO-5 Well-Being Index

## Abstract

Nurses’ well-being has been increasingly recognised due to the ongoing pandemic. However, no validation scales measuring nurses’ well-being currently exist. Thus, we aimed to validate the WHO-5 Well-Being Index (WHO-5) in a sample of nurses. A cross-sectional multinational study was conducted, and a total of 678 nurses who worked during the COVID-19 pandemic in Spain (36.9%), Chile (40.0%) and Norway (23.1%) participated in this study. The nurses completed the WHO-5, the Patient Health Questionnaire-2 (PHQ-2), the Generalized Anxiety Disorder-2 (GAD-2) and three questions about the quality of life (QoL). The WHO-5 demonstrated good reliability and validity in the three countries. Cronbach’s alphas ranged from 0.81 to 0.90. High correlations were found between the WHO-5 and the psychological well-being dimension of QoL, and negative correlations between the WHO-5 and PHQ-2. The unidimensional scale structure was confirmed in all the countries, explaining more than 68% of the variance. The item response theory likelihood ratio model did not show discernible differences in the WHO-5 across the countries. To conclude, the WHO-5 is a psychometrically sound scale for measuring nurses’ well-being during a pandemic. The scale showed strong construct validity for cross-cultural comparisons; however, more research is required with larger sample sizes.

## 1. Introduction

Psychological well-being is a relative rather than an absolute concept [1]. Thus, one’s satisfaction or happiness is influenced by a blend of objective reality and one’s subjective reactions to it. This is affected by how well a person functions and how much their aspirations differ from their current situation [1,2]. As a part of the quality of life (QoL) concept, well-being is measured via diverse self-rated scales, although no consensus exists on a gold standard [3,4]. However, in recent years, collecting self-rated well-being data has become valuable for understanding well-being among the general population [5], in research [6,7], in clinical settings [7] and for understanding healthcare worker-reported outcomes in the workplace [5,8].

Nursing professionals frequently face burdensome workloads and are exposed to severe emotional demands and high perceived stress levels [9]. This is further complicated by the likelihood of understaffing and working during stressful events, such as a global pandemic. Furthermore, having to care for patients in an unsafe environment while exposed to personal risk can negatively affect nurses’ physical health [10] and psychological well-being [10,11]. In addition to a potential reduction in work performance, these factors can be associated with anxiety and depression symptoms [12,13] with deterioration of psychological well-being [11,14]. Nurses’ well-being is a significant determinant of a hospital’s ability to provide patient care [9]. As significant concerns for nurses’ well-being have been documented during the current global pandemic [5,11,15]; it is crucial to have a reliable self-reported scale that can adequately assess the well-being of nursing professionals. Thus, a critical nursing research goal is to capture and improve nurses’ well-being worldwide, and the WHO-5 Well-Being Index (WHO-5) is potentially useful for measuring this [7].

The WHO-5 reflects positive affections, and this short scale captures subjective psychological well-being by measuring affective and hedonic dimensions of well-being [2,16]. This self-rated scale was developed to enable primary healthcare general practitioners to screen patients for signs of depression to reduce relapse of depression symptoms and suicide rates [17,18]. Since its development, the scale has been used to guide clinical practice worldwide [7]. As a generic self-rated well-being scale, previous research on its psychometric properties has focused primarily on students and patients in different settings, such as general practice, private clinics, and hospital settings [6,7]. In addition, it is increasingly used as a reliable measure to monitor depression [19], with excellent clinimetric cross-cultural sensitivity to detect depression [20]. Extensive research and translations into the languages of over 30 countries are among the advantages of the WHO-5, allowing the understanding of mental health from a well-being perspective. However, although previous work has reported adequate internal consistency and structural validity of the scale in several research fields [1,7], no cross-cultural validation scales measuring nurses’ well-being currently exist.

When considering using a previously developed self-rated scale, examining validation and adaptation to specific context conditions of the population is recommended to ensure that the scale measures the originally intended construct [21]. Hence, researchers can determine the items’ suitability to capture the construct in advance, increasing the certainty that the scale would provide reliable information. This would give the information required to justify using the scale if appropriate for the specific new population. Although the WHO-5 was first validated as a scale for psychological well-being among patients, other factors can affect the scale’s psychometric properties. These factors could include cultural and socioeconomic background [22], and other factors that can highlight whether the hypothesised factor structure is the same across groups [21]. A recent validation study using item response theory (IRT) suggested that cross-cultural validation studies are needed for the WHO-5 [23]. This recommendation is consistent with recent studies suggesting that the cross-cultural validity of well-being scales remains an unexplored question [24]. These recommendations align with Boer et al. [25], suggesting that comparisons between countries may require assurance of measurement comparability before reliable conclusions can be drawn. However, despite previous use of the WHO-5 Well-Being Index in nursing studies [26], measurement invariance investigations of the WHO-5 have yet to be conducted among nurses. Furthermore, evidence-based psychometric evaluations in a pandemic context are relatively scarce.

The present study aimed to investigate the validity of the WHO-5 Well-Being Index among clinical nurses working in health services in Spain, Chile and Norway. This was performed by examining the internal consistency and conducting factor and IRT analyses, including assessing measurement invariance and differential item functioning (DIF) across countries and comparing the results obtained. Several a priori research hypotheses were tested to investigate the internal structure of the WHO-5. First, it was hypothesised that the WHO-5 would have good internal consistency, with Cronbach’s alpha coefficients exceeding 0.75. Second, as previous research has suggested that subjective well-being is part of the concept of perceived QoL [27], we also expected a priori that the WHO-5 would show high and strong correlations with questions about QoL. As validation studies in recent years have suggested that psychological problems affect subjective well-being [28,29], we also expected that core anxiety symptoms would have a negative association with the WHO-5 and that the WHO-5 would be negatively associated with core symptoms of depression. Finally, it was hypothesised that the WHO-5 would have adequate structural validity and a good fit in a one-factor solution based on prior studies [7].

## 2. Materials and Methods

This cross-sectional multinational study with nonprobability snowball sampling was conducted following the guidelines of Consensus-Based Standards for the Selection of Health Measurement Instruments [30]. An online survey was used to collect the data, and the STROBE reporting guideline was followed.

### 2.1. Sample/Participants

The recruitment of nurses followed three steps. First, schools’ management, nurse educators and the teaching staff at three universities invited nurses working in non-emergency clinical settings, including university hospitals, non-teaching hospitals and public community local health settings. Second, the survey was posted on the web pages of the associations of nurses. Finally, nurses connected to the university campus and potential participants were kindly asked to invite other nurses by forwarding the information about the study to other networks and via social media. Data were obtained during June and July 2020. Nurses were recruited to participate in the online survey by sending a welcoming email containing a hyperlink to the survey to all the nurses registered in professional associations in the areas of study (Tenerife Island in Spain, Santiago in Chile and Mid and West Norway).

Inclusion criteria were direct care nursing staff, working in inpatient wards, and employed by the hospitals. In the survey, questions about current working position were included, that is, whether they were involved with clinical, academic or administrative work, or were working with nursing education at the university. Participants were included if they were nurses, actively providing direct patient care in their respective countries, and agreed to participate.

### 2.2. Data Collection

We asked the participants to anonymously provide their sociodemographic information, through self-reporting, as part of the different questionnaires in the online survey. All the items were set as voluntary. To ensure that participants did not answer repeatedly, the online survey was set to reject multiple responses from the same IP address. We designed the survey to avoid burden on the respondents, maximise data quality and maintain ethically sound research. In addition, the online survey was designed so it was easy for the nurses to navigate. It was planned a priori to calculate item response rates and to exclude participants omitting answers/missing (if 25% or fewer items were missing). However, all participants answered the WHO-5 completely without omissions.

Before carrying out the investigation, the survey was piloted by five nurses and two professors. They examined the content validity of the items, reviewed their relevance, and provided feedback to ensure language representation for ease of understanding.

### 2.3. Measures

#### 2.3.1. WHO-5 Well-Being Index

The nurses self-reported their well-being during the past two weeks. The scale has five items depicting feeling cheerful (Item 1: ‘I have felt cheerful and in good spirits’), feeling calm (Item 2: ‘I have felt calm and relaxed’), feeling active (Item 3: ‘I have felt active and vigorous’), feeling rested when waking up (Item 4: ‘I woke up feeling fresh and rested’) and feeling that one’s life is filled with exciting things (Item 5: ‘My daily life has been filled with things that interest me’). The response options ranged from 0 to 5, with 0 representing ‘at no time’ and 5 ‘all the time’ [6,7]. In the present study, the WHO-5 Well-Being Index was calculated as the sum of the scores of the responses, ranging from 0 (the worst imaginable well-being) to 25 (the best imaginable well-being). We used the Spanish version of the WHO-5, and this version was validated and tested for clarity in a sample of outpatients in community mental health settings [31]. The Norwegian version we used was validated among adolescents [32].

#### 2.3.2. Quality of Life

The nurses’ self-reported QoL was assessed using the Multidimensional Quality of Life Index (MQLI). The MQLI is a self-administered questionnaire. The items evaluated their physical well-being, psychological/emotional well-being and overall QoL on a 10-point line [27,33]. Responses to these three questions were scored from 0 (representing ‘poor’) to 10 (indicating ‘excellent’) by placing a mark on the value representing their experiences. The reliability and validity of the MQLI were established in the original validation study (Cronbach’s alpha 0.92) and in the Norwegian validation study (Cronbach’s alpha 0.73) [27,33]; however, investigations about the validity and reliability of the MQLI-3 items have not been conducted among nurses.

#### 2.3.3. Anxiety and Depression

The nurses’ self-reported anxiety was assessed using the Generalized Anxiety Disorder 2 (GAD-2) scale during the past two weeks [34]. In the present study, the GAD-2 was used as a self- administered questionnaire, and nurses were asked to report the presence of each symptom during the last 14 days. The questionnaire assessed how often they have been nervous (‘feeling nervous, anxious or on edge’) and worried (‘not being able to stop or control worrying’). The responses were provided on a Likert scale, from 0 (representing ‘not at all’) to 3 (representing ‘nearly every day’). The construct validity of the PHQ-2 was established in the original validation study [34].

The nurses’ self-reported depression was assessed using the Patient Health Questionnaire-2 (PHQ-2) scale during the past two weeks [35]. In the present study, the PHQ-2 was used as a self-administered questionnaire. The questionnaire asks how often a person has been bothered by feeling a lack of interest (‘little interest or pleasure in doing things’) and feeling sad (‘feeling down, depressed, or hopeless’). Nurses were asked to report the presence of each symptom during the last 14 days. Responses are provided on a Likert scale between 0 and 3, where 0 represents ‘not at all’, 1 represents ‘several days’, 2 represents ‘more than half the days’, and 3 represents ‘nearly every day’. The PHQ-2 is widely used and considered to be a reliable measure. The construct and criterion validity of the PHQ-2 were established in the original validation study [35].

### 2.4. Data Analysis

The data were analysed using IBM SPSS Statistics for Macintosh, Version 25.0 (IBM Corp., Armonk, NY, USA), and R library psych with ULLRToolbox. Values were described with means, standard deviations, frequencies and percentages. Each survey was examined for completeness, floor, and ceiling effects. Internal consistency was calculated using Cronbach’s alpha, and factor analyses were performed using principal component and minimum rank analysis. Two studies were conducted to assess the WHO-5 Well-Being Index invariance through the three samples (countries). First, an analysis was generated from IRT to study the consistency of the scale in a situation of invariance. A DIF study was conducted to assess item stability across different samples and test a potential source of systematic measurement bias in item responses regarding culture. The likelihood ratio chi-square test, the Nagelkerke and McFadden’s pseudo-R were computed as magnitude measures with a minimum cell count of six. The study of the differential behaviour of the item allowed us to study the invariance of the construct through different groups of participants [36].

Simultaneously, a study of the construct invariance was conducted through a confirmatory factor analysis in which configurational and metric invariance across groups were tested [37,38]. The IRT analysis was conducted following a logistic ordinal regression differential item functioning model under invariance criteria [39], using the country variable as a grouping variable. Three search criteria for items were used according to the country (Spain, Chile, and Norway): chi-square, R^2^ and beta. None of the cases was flagged. Our aim was to verify the linear item invariance [40] comprising the construct according to the country variable. The proposed models were compared following Satorra and Bentler approach [41].

Sample size estimates were based on factor analysis, requiring at least 10 participants per variable to achieve replicable findings, following the guidelines of Consensus-Based Standards for the Selection of Health Measurement Instruments [30].

### 2.5. Ethical Approval

The Research Ethics Committee of the Canary Islands Health Service, Spain, first approved the study (CHUC_2020_33), and subsequently, we obtained approvals in Chile and Norway (27/2020 and 155172). All the participants provided informed consent before participation.

## 3. Results

Completed data were obtained from 678 nurses. Their mean age was 39.3 years (standard deviation (SD) = 12.1), ranging from 36 to 48 years, and most nurses were female (74.5% for Chile, 88.9% for Norway, and 80.8% for Spain). There were no missing values. None of the items have floor/ceiling effects. The mean scores for each of the measures used in this study are shown in Table 1.

The mean WHO-5 score for the global sample was 13.0 (SD = 5.1), the median score was 13.0, and the skewness was −0.17 (SE = 0.94). In the samples in Chile, Norway, and Spain, the WHO-5 items registered high correlations with each other (Chile: from 0.56 to 0.75; Norway: from 0.31 to 0.51; Spain: from 0.59 to 0.75), indicating that they measured the same construct. Similarly, the WHO-5 items showed a high correlation with the total scale score (Chile: from 0.70 to 0.82, Norway: from 0.53 to 0.73; and Spain: from 0.64 to 0.78).

Table 1 also shows the results regarding the hypotheses and correlations. Strong correlations were expected based on the assumption that the WHO-5 Well-Being Index is part of the concept of perceived QoL, measured using the MQLI-3. These were supported by the WHO-5 Well-Being Index and physical well-being (*r* = 0.662 for Chile, *r* = 0.683 for Norway, and *r* = 0.592 for Spain), psychological/emotional well-being (*r* = 0.738 for Chile, *r* = 0.610 for Norway, and *r* = 0.721 for Spain), and overall QoL (*r* = 0.724 for Chile, *r* = 0.584 for Norway, and *r* = 0.630 for Spain).

Based on previous studies, we expected that the WHO-5 Well-Being Index is a valid measure in the context of mental health, and strong correlations were expected with core symptoms of anxiety (GAD-2) and depression (PHQ-2). Table 1 shows adequate negative correlations obtained with GAD-2 (*r* ranging from −0.39 to −0.77) and PHQ-2 (*r* ranging from −0.56 to −0.73).

The WHO-5 Well-Being Index showed high internal consistency in the three countries (Table 2, Chile-Cronbach’s alpha = 0.903; Spain-Cronbach’s alpha = 0.883; and Norway-Cronbach’s alpha = 0.810). No item deletion improved Cronbach’s alpha in any of the samples studied.

The unidimensionality of the WHO-5 Well-Being Index was confirmed through factor analysis using principal component analysis as an extraction method. A single-factor structure was demonstrated, explaining 72.5% in Chile (factor loadings between 0.80 and 0.90), 57.7% in Norway (factor loadings between 0.70 and 0.84) and 68.7% of the variance in Spain (factor loadings between 0.77 and 0.87).

Table 2 shows the factor loadings. The Kaiser–Meyer–Olkin measures of sampling adequacy were 0.859 for Chile, 0.793 for Norway and 0.846 for Spain, indicating sample adequacy. Bartlett’s test of sphericity values were 672.275 (Spain), 859.431 (Chile) and 257.384 (Norway; df = 10, *p* < 0.0001), all of them indicating an underlying structure in the scale and that factor analyses were justified in the samples.

The comparison data of both models can be seen in Table 3. As shown in Figure 1, a broad overlap was found between the Spain and Chile distributions, although the Norway sample showed a higher mean well-being score than Spain and Chile samples. With this setting in two iterations, three items were identified (flagged) as potential sources of differences regarding country: Item 1 (‘I have felt cheerful and in good spirits’), Item 3 (‘I have felt active and vigorous’) and Item 4 (‘I woke up feeling fresh and rested’; see Figure 1).

Figure 2 shows the item true score function with a test for differential item functioning models (uniform vs. nonuniform) and item response function for Items 1, 3 and 4 with regression parameter values by country, indicating that the items with differential behaviour depending on the country were Items 1, 3 and 4. A differential effect was found in response to Item 1 depending on the type of country [Pr(
$\chi_{12}^{2}$
,2) < 0.001]. The slope for the Norwegian sample was lower than that of the Spanish and Chilean samples (1.95 vs. 3.71 and 3.35). Data found in the item true score function showed that the responses of the Norwegian sample for this item were in a low–medium range of the trait compared to the Spanish and Chileans (Figure 2b). Item 3 showed a differential effect of the item [Pr(
$\chi_{12}^{2}$
,2) < 0.001]. Differential behaviour was evident: the second graph of true item scores for Item 3 showed greater homogeneity between the three samples. Finally, Item 4, ‘I have felt cheerful and in good spirits’, also showed differential behaviour according to the country [Pr(
$\chi_{12}^{2}$
,2) < 0.001]. Figure 2f shows the trait values according to the true response function, where the Norwegian sample showed higher trait values than the Spanish and Chilean samples.

The values of the differential-corrected and differential-uncorrected raw data per participant and group regarding the central 50% of the distribution showed an interquartile range between −0.04 and 0.02 with a median value of approximately 0.0. The differential behaviour of the scale according to the countries showed some spurious differences between the raw values and values according to the differential. The IRT differential item analysis showed a spurious bias response effect regarding Items 1, 3 and 4. We then considered whether some items exhibiting differential behaviour affected the internal validity of the scale structure according to the countries. Our aim was to verify the linear item invariance comprising the construct according to the country variable. The comparison of the configurational invariance, assuming the same measurement model per country, and the invariance in the beta regression coefficients (per country) were significant [χ2 diff(4) = 23, *p* < 0.001)]. A statistical significance was found, which reflected as a small effect size (w = 0.01). It was assumed that factorial weights differed by country. This spurious difference (located in the weights of Items 1, 3 and 4) was lower on average in the Norwegian sample (see Figure 2).

## 4. Discussion

This study aimed to investigate the validity of the WHO-5 Well-Being Index for nurses by examining internal consistency and conducting factor and IRT analyses. This was conducted to investigate the WHO-5’s scale construct validity across three countries: Spain, Chile, and Norway.

Construct validity based on a priori hypothesis testing was supported. Contrary to previous studies showing low Cronbach’s alpha [42], the WHO-5 Well-Being Index showed high internal consistency in the three countries, with Cronbach’s alpha varying from 0.810 to 0.903 (Norway to Chile). These findings are within the range of Cronbach’s alpha reported in validation studies among medical educators in Hong Kong [43], outpatients with epilepsy in Denmark [44], adults living with epilepsy and HIV in Kenya [45] and among Chinese university students [46].

Regarding hypotheses concerning well-being and QoL, our study revealed high correlation values between the WHO-5 Well-Being Index and the psychological/emotional well-being dimension of the MQLI and the overall QoL. These correlations are consistent with those Mundal et al. [27] reported in a previous validation study. Our results also indicated adequate negative correlations between well-being and anxiety, consistent with a previous study that found strong correlations between the WHO-5 Well-Being Index and anxiety [29]. Additionally, adequate negative correlations were found between well-being and depression, consistent with previous studies reporting that the WHO-5 correlated negatively with depression [28,47,48,49,50,51,52]. Together with the obtained internal consistency values, these findings support the construct validity of the well-being scale.

The unidimensionality of the WHO-5 was confirmed through factor analysis, with values from 57.7% (for the Norwegian sample) to 72.5% (for the Chilean sample). Our findings are consistent with prior research founding a one-factor structure [7,52], indicating that the WHO-5 can be used to measure the well-being of nurses in different countries for cross-cultural investigations. Notably, as IRT DIF analysis revealed, the WHO-5 performed differently for the well-being construct in the Norwegian sample than for the Spanish and Chilean ones. However, the differential patterns found were associated with negligible effect sizes below 0.13 [53]. Additionally, the Norwegian nurses had a greater probability of responding with high values on the scale than the Spanish and Chilean. This suggests that these items behave slightly differently depending on the country. Cultural differences are likely to have caused the differences in responses to the items. When measuring well-being in countries with different economic situations [22], for instance, developed countries such as Norway compared to other countries such as Chile, this difference in economic well-being could also be a source of bias.

It is noteworthy that differences between the samples could be explained by the impact of COVID-19, since Chile and Spain were hit much harder than Norway, specifically during the data collection period. As prior studies have suggested [54,55,56], factors such as self-perceived job insecurity, ethical dilemmas and stringency of government responses may affect well-being. Nevertheless, nurses in the Norwegian sample reported higher values when comparing the means between the three countries. These combined findings emphasize that nurses from Norway reported better well-being than those in Spain and Chile during the pandemic. These results appear consistent with studies reporting that the WHO-5 Well-Being Index can differentiate between populations [31,43,45]. However, further research, using several measurement points, is needed to investigate whether the nurses’ mental well-being changes over time and whether the WHO-5 Well-Being Index can capture these changes.

Although it is unknown whether the pandemic might explain these differences or whether the differences might be explained by language differences in how nurses perceived their well-being, it is noteworthy that the overall mean of the WHO-5 was significantly lower for nurses from Chile and Spain, as this was higher compared to the WHO-5 means reported in a similar COVID-19 study in Vietnam [57]. Nurses in Chile and Spain had mean scores below 13, corresponding to depression. Our findings regarding the lack of well-being reported in all the countries have implications for government policies/policymakers, showing that they should focus on nurses’ well-being. Additionally, nurses in the three countries reported variations in their well-being with a scale that also detects depression [7]. Our findings suggest that the COVID-19 pandemic influenced how nurses struggled with feeling ‘calm and relaxed’ and ‘active and vigorous’ and waking up ‘feeling fresh and rested’. Such information will guide researchers seeking interventions to enhance well-being in different cultures.

Notably, nurses in Chile and Spain, countries hit hard by COVID-19, struggle more with mental health problems. This finding is in line with a prior study [58]. In countries where depression is associated with stigma [59], the WHO-5 Well-Being Index will be a better alternative to measure depression and lack of well-being. Thus, measuring nurses’ well-being can guide hospital administrators in implementing strategies to protect nursing staff without time constraints and psychological burdens being viewed as a stigma. In addition, collecting such information in different countries can guide researchers in developing methods to improve nurses’ well-being across countries and cultures.

Although the study’s multi-country design, involving collecting data in three countries, and rigorous statistical analyses are strengths, the study has limitations. First, the nurses studied were convenience-sampled, using a non-probability snowball sampling approach, limiting the possibility to know the response rate of nurses who refused to participate in the study. Second, the sample is limited, as demographic differences existed between the countries. Finally, cultural differences may have caused the differences regarding Item 1, as our result suggested that this item could be understood differently in different countries. Nevertheless, while accepting this final limitation, it must be noted that when measuring well-being in countries with different economic situations, such as developed countries such as Norway as compared to countries such as Chile, this difference in economic well-being could be a source of bias. However, further studies are required due to the relatively limited sample size.

## 5. Conclusions

The WHO-5 Well-Being Index demonstrated its utility as a cross-cultural ultra-brief questionnaire for measuring subjective psychological well-being in Spanish, Chilean and Norwegian nurses. The scale showed high internal consistency in the three countries. Although the unidimensionality of the WHO-5 was confirmed through factor analysis, we found a non-invariance effect on the weights of items, and Item 1 appears somewhat less stable when comparing the Norwegian sample with the Spanish and Chilean samples. Although our findings support the scale’s construct validity, allowing comparative analyses between countries, more research is required with larger sample sizes.

## Figures and Tables

**Figure 1 ijerph-19-10106-f001:**
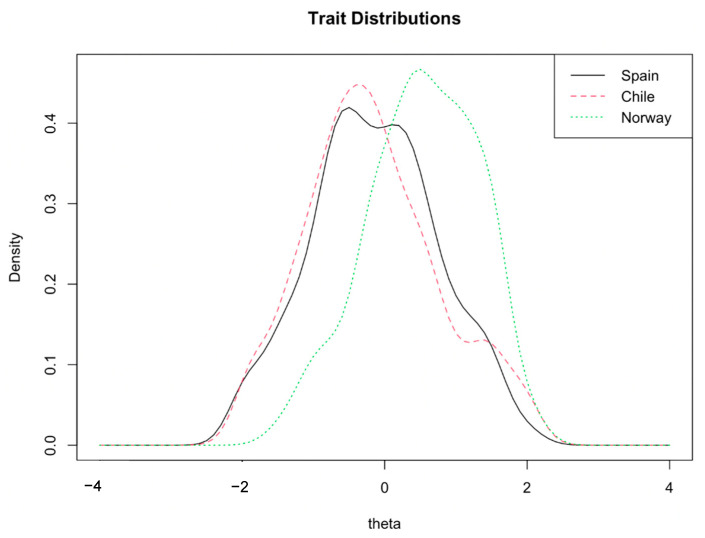
Graphic representation of the population latent trait distribution in the three countries.

**Figure 2 ijerph-19-10106-f002:**
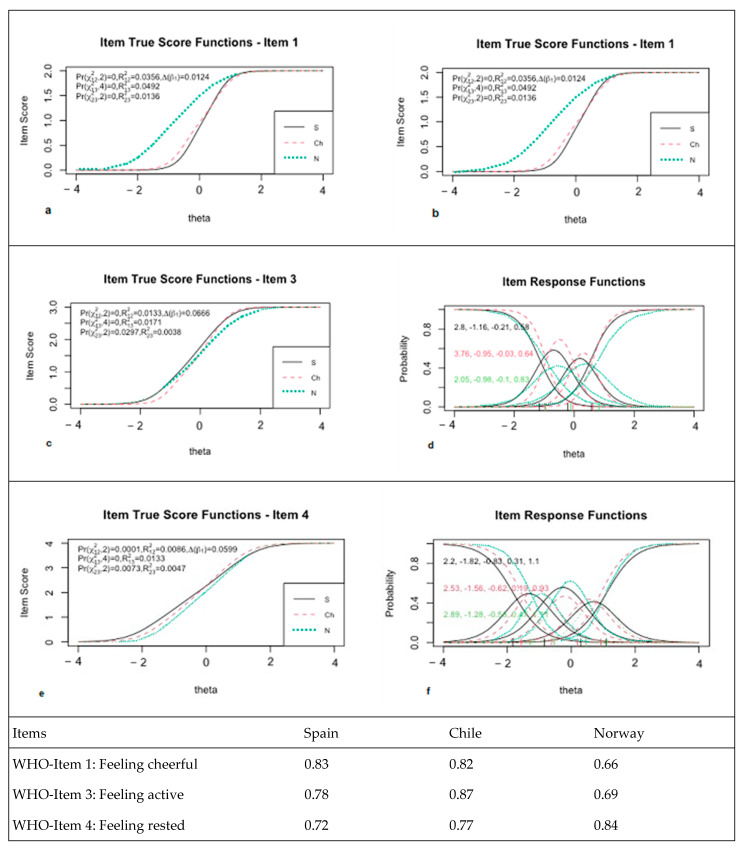
Item true score function with test for differential items functioning models (uniform vs. nonuniform) and item response function for Items 1 (**a**,**b**), 3 (**c**,**d**) and 4 (**e**,**f**) with regressions parameters values by country.

**Table 1 ijerph-19-10106-t001:** Means (M), standard deviation (SD, ±) and bivariate correlations of the study variables.

Variables	Spain (*n* = 250)	Chile (*n* = 271)	Norway (*n* = 157)
WHO-5 Total Score, mean, SD ***	12.4 ± 4.9	11.9 ± 5.3	16.0 ± 3.7
MQLI-physical well-being, mean, SD ***	6.5 ± 2.3	6.1 ± 2.4	6.5 ± 2.1
MQLI-emotional well-being, mean, SD ***	6.2 ± 2.5	5.8 ± 2.7	7.0 ± 1.9
MQLI-quality of life, mean, SD ***	6.9 ± 2.4	6.4 ± 2.5	7.6 ± 1.8
GAD-2, mean, SD ***	2.5 ± 1.7	2.9 ± 1.8	1.2 ± 1.1
PHQ-2, mean, SD ***	2.0 ± 1.7	2.2 ± 1.7	1.1 ± 1.0
Correlations MQLI-physical well-being and WHO-5	0.592 **	0.662 **	0.683 **
Correlations MQLI-emotional well-being and WHO-5	0.721 **	0.738 **	0.610 **
Correlations MQLI-quality of life and WHO-5	0.630 **	0.724 ***	0.584 **
Correlations GAD-2 and WHO-5	−0.650 **	−0.733 **	−0.390 **
Correlations PHQ-2 and WHO-5	−0.726 **	−0.698 **	−0.563 **

** *p* < 0.01. *** *p* < 0.001. GAD-2 = Generalized Anxiety Disorder-2; MQLI = Multidimensional Quality of Life Index; PHQ-2 = Patient Health Questionnaire-2; WHO-5 = WHO-5 Well-Being Index.

**Table 2 ijerph-19-10106-t002:** Means (M), standard deviation (SD, ±), factor loading and Cronbach’s alpha for the three samples.

WHO-5 Item	Spain (*n* = 250)		Chile (*n* = 271)		Norway (*n* = 154)	
	Mean ± SD	Factor Loading	Mean ± SD	Factor Loading	Mean ± SD	Factor Loading
WHO-Item 1: Feeling cheerful	2.7 ± 1.1	0.869	2.7 ± 1.1	0.853	3.7 ± 0.6	0.699
WHO-Item 2: Feeling calm	2.4 ± 1.1	0.867	2.2 ± 1.2	0.874	3.3 ± 0.9	0.725
WHO-Item 3: Feeling active	2.6 ± 1.2	0.843	2.4 ± 1.2	0.896	3.0 ± 1.1	0.730
WHO-Item 4: Feeling rested	2.2 ± 1.2	0.794	2.1 ± 1.3	0.832	2.6 ± 1.2	0.843
WHO-Item 5: Feeling that one’s life is filled with interesting things	2.5 ± 1.3	0.768	2.5 ± 1.3	0.801	3.4 ± 0.9	0.792
Cronbach’s alpha	0.883		0.903		0.810	

**Table 3 ijerph-19-10106-t003:** Comparison data between models.

Models	Df	AIC	BIC	Chi-Square	Chi-Square Diff	Df Diff
Model 1	15	8771.5	8907.9	82.918		
Model 2	23	8801.7	8800.3	128.376	53.445	8 ***

*** *p* < 0.001. AIC = Akaike Information Criteria; BIC = Bayes Information Criteria; Df diff = Degree of Freedom differential analysis.

## Data Availability

Due to the sensitive nature of the questions asked in this study, survey respondents were assured raw data would remain confidential and would not be shared. The data are not publicly available due to data protection regulations.

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
