# Peer review of "Psychometric Properties of the WHO-5 Well-Being Index among Nurses during the COVID-19 Pandemic: A Cross-Sectional Study in Three Countries"

_ijerph, 2022, doi:10.3390/ijerph191610106_

Round 1

Reviewer 1 Report

The authors present a study with 678 nurses in 3 countries. The WHO-5 Well-Being-Index was validated in this sample size of nurses. The topic ist very interesting and "up to date". The manuscript is worth publishing, some minor corrections should be performed.

Material and methods, Sample: This is the only weak part of the draft. How many nurses were approached? (Drop-out-rate?) Were they from one university hospital or other academic hospital? What kind of nurses: emergency, geriatric, internal wards?

The authors should explain and answer these questions. 

Author Response

Dear Review #1

Thank you for taking the time to evaluate our work. We appreciate your input and have revised the manuscript to comply with your suggestions. Below we have addressed your comments, point by point, and have made changes to the manuscript accordingly (pages 3 and 11).

Comment and suggestions for authors Material and methods, Sample: This is the only weak part of the draft. How many nurses were approached? (Drop-out-rate?) Were they from one university hospital or other academic hospital? What kind of nurses: emergency, geriatric, internal wards?

            Response: Thank you for pointing this out. Unfortunately, although we agree that information about drop-out rate is important, we have no possibility to calculate this. Nevertheless, to facilitate clarity and better justify our position, some information has been added to the Method section about the settings and what kind of nurses were enrolled (page 3). We have also included your remark in the limitations section (page 11).

Reviewer 2 Report

Dear authors,

The manuscript entitled "Psychometric properties of the WHO-5 Well- Being Index among nurses during the COVID-19 pandemic: A cross-sectoral study in three countries" refers to

 A cross-sectional study in three countries aiming to validate the WHO-5 Well-Being Index (WHO-5) in a sample of nurses.

Health professionals, namely nurses, as the authors point out, are subject to activities that overload them both physically and emotionally, because it is actually necessary to create instruments or, if they already exist, they must be translated and culturally adapted. The assessment instruments allow a quick and objective assessment of the situation, allowing a situational diagnosis and later the adoption of monitored corrective measures.

The abstract is adequately structured, answering all the questions necessary to understand the investigation carried out.

In the introduction, all the concepts under analysis are addressed. In line 100 the authors present a reference according to the. APA standards, so they must change to the journal's standards.

Materials and Methods: Who translated and cross-culturally adapted the WHO-5 scale for the three countries included? Who validated your content? Did they turn to experts? If you have already published an article on these issues, please refer to these bibliographic references.

Discussion: In line 317 the authors present a reference according to the APA Norms, so they should proceed with the change to the journal's norms.

They use current and relevant bibliographic references both in the introduction and in the discussion of the results.

Congratulations and good work!

Author Response

Reviewer #2:

Thank you for your kind words and for providing us with this insightful review of our work. Below we have addressed your comments, point by point. In addition, corresponding changes are highlighted in the revised manuscript (with the modified text in red).

Comment 1: In the introduction, all the concepts under analysis are addressed. In line 100 the authors present a reference according to the. APA standards, so they must change to the journal's standards.

Response 1: Thank you, this reference has been rectified.

Comment 2, Materials and Methods: Who translated and cross-culturally adapted the WHO-5 scale for the three countries included? Who validated your content? Did they turn to experts? If you have already published an article on these issues, please refer to these bibliographic references.

Response 2: We agree that this was unclear. Thank you for pointing this out. The Norwegian version of the scale was validated among adolescents and was translated by Christensen and colleagues (a reference has been added). We have recently submitted a manuscript describing the validation of the WHO-5, but since the manuscript is under review (manuscript ID ijerph-1850271), we will not include this information. However, since our team was adapted the WHO-5, and the Spanish validation study has been published (a reference has been added in page 4). Additionally, as in our study five nurses and two professors examined the content validity of the initial items and they checked language representation for ease of understanding; we have added this information on page 4.

Comment 3, Discussion: In line 317 the authors present a reference according to the APA Norms, so they should proceed with the change to the journal's norms.

Response: Thank you, the APA-reference you are referring to has been rectified.

Reviewer 3 Report

Nursing professionals often face heavy workloads and face severe emotional demands and high perceived stress levels. In addition, having to care for patients in an unsafe environment while facing personal risks can negatively impact nurses' physical and mental health. The situation is further complicated by the potential for understaffing and work during stressful events such as a global pandemic. In addition to a potential reduction in job performance, these factors may also be associated with worsening anxiety and depressive symptoms and mental health. Nurses' well-being is an important determinant of a hospital's ability to provide patient care.

The topic is very important, and thanks for the opportunity to review it.

I have reviewed the manuscript. The present study aimed to investigate the validity of the WHO-5 Well-Being Index among clinical nurses working in health services in Spain, Chile and Norway. The present study shows that the WHO-5 Well-Being Index demonstrated its utility as a cross-cultural ultra-brief questionnaire for measuring subjective psychological well-being in Spanish, Chilean and Norwegian nurses.

However, there are some concerns.

1. Collecting data through online anonymous surveys, how can researchers ensure that participants answer repeatedly or even multiple times?

2. Collecting data through online surveys, how does the researcher ensure that all the respondents answering the questions are answered completely without omissions? If there are omissions, how will the researcher deal with the question?

3. Collecting data through online surveys, how does the researcher ensure that participants are not falsely answering, such as ticking the same score for all items?

4. In the subsection of Measures, in addition to explaining the answering principles of each questionnaire, please also provide information on the reliability and validity of each questionnaire.

Author Response

Reviewer #3:

We gratefully thank you for the precious time that you spent making constructive remarks and valuable suggestions. Your input has been addressed in this revision. Corresponding changes are highlighted in the revised manuscript (with the modified text in red, on pages 3 and 4).

Comment 1: Collecting data through online anonymous surveys, how can researchers ensure that participants answer repeatedly or even multiple times?

Response 1: To ensure that participants did not answer repeatedly, the survey was set to reject multiple responses from the same IP address. We have added a comment on this on page 3.

Comment 2: Collecting data through online surveys, how does the researcher ensure that all the respondents answering the questions are answered completely without omissions? If there are omissions, how will the researcher deal with the question?

Response 2: We designed the survey to avoid the respondent burden, maximise data quality and maintain ethically sound research; all items were set as voluntary. In addition, the online survey was designed so it was easy for the nurses to navigate. It was planned a priori to calculate item response rates and to exclude participants omitting answers/missing (if 25% or fewer items were missing). However, there were no missing values, and we have added this information on pages 3 and 4.

Comment 3: Collecting data through online surveys, how does the researcher ensure that participants are not falsely answering, such as ticking the same score for all items?

Response 3: To control for this bias, we balanced the survey questions by including other scales (positive and negative options), checked correlations and extreme responses, and designed the survey to keep participants focused, avoiding using one type of question, mixing binary and different types of Likert questionnaires. In addition, we tried to reduce response bias by making our survey anonymous. When a survey is anonymous, respondents are more inclined to discuss sensitive issues and provide more detailed and honest feedback.

Comment 4: In the subsection of Measures, in addition to explaining the answering principles of each questionnaire, please also provide information on the reliability and validity of each questionnaire.

Response 4: Thank you for the suggestions. We agree with the reviewer and apologise for being unclear regarding the validity and reliability of all included questionnaires. In this revision, we have added more information, explaining each questionnaire’s answering principles, reliability, and validity (page 4).
